# Structural basis for ligand and innate immunity factor uptake by the trypanosome haptoglobin-haemoglobin receptor

Harriet Lane-Serff[1], Paula MacGregor[2], Edward D Lowe[1], Mark Carrington[2]*, Matthew K Higgins[1]*

[1]Department of Biochemistry, University of Oxford, Oxford, United Kingdom; [2]Department of Biochemistry, University of Cambridge, Cambridge, United Kingdom

**Abstract** The haptoglobin-haemoglobin receptor (HpHbR) of African trypanosomes allows acquisition of haem and provides an uptake route for trypanolytic factor-1, a mediator of innate immunity against trypanosome infection. In this study, we report the structure of *Trypanosoma brucei* HpHbR in complex with human haptoglobin-haemoglobin (HpHb), revealing an elongated ligand-binding site that extends along its membrane distal half. This contacts haptoglobin and the β-subunit of haemoglobin, showing how the receptor selectively binds HpHb over individual components. Lateral mobility of the glycosylphosphatidylinositol-anchored HpHbR, and a ~50° kink in the receptor, allows two receptors to simultaneously bind one HpHb dimer. Indeed, trypanosomes take up dimeric HpHb at significantly lower concentrations than monomeric HpHb, due to increased ligand avidity that comes from bivalent binding. The structure therefore reveals the molecular basis for ligand and innate immunity factor uptake by trypanosomes and identifies adaptations that allow efficient ligand uptake in the context of the complex trypanosome cell surface.

*For correspondence: mc115@cam.ac.uk (MC); matthew.higgins@bioch.ox.ac.uk (MKH)

**Competing interests:** The authors declare that no competing interests exist.

## Introduction

African Animal Trypanosomiasis is one of the major constraints on the productivity of pastoralists in sub-Saharan Africa and can be caused by infection by a range of trypanosome species (*Shaw, 2004*), while infections of humans are caused by only two subspecies of *Trypanosoma brucei* (*Laveran, 1902*; *Pays and Vanhollebeke, 2009*). The disease is persistent as the host immune system is usually unable to clear the infection. This is due to the trypanosome having evolved a population survival strategy based on autoregulation of parasitaemia and antigenic variation (*MacGregor et al., 2011*; *Horn, 2014*). The trypanosomes also internalize and degrade surface bound immunoglobulin (*Pal et al., 2003*; *Engstler et al., 2007*), increasing the survival of an individual cell and thereby increasing the likelihood of transmission. Both of these strategies require a densely packed cell surface coat of variant surface glycoprotein (VSG) that acts as a barrier, preventing access of host immunoglobulins to the plasma membrane (*Schwede and Carrington, 2010*). This coat also undergoes antigenic variation through expression of a single VSG gene from a genomic repertoire of hundreds (*Horn, 2014*).

Although the VSG coat restricts immunoglobulin access, it must be permissive for receptor-mediated binding and uptake of macromolecular ligands. *T. brucei*, and the closely related *T. congolense*, have receptors for both transferrin (TfR) for iron (*Steverding et al., 1994*; *Schell et al., 1991*; *Jackson et al., 2013*) and haptoglobin-haemoglobin (HpHbR) for haem (*Vanhollebeke et al., 2008*; *Higgins et al., 2013*). These are held on the external face of the plasma membrane by covalent attachment of the C-terminal carboxyl group to a glycosylphosphatidyl inositol to form a GPI-anchor. All have free movement in the lateral plane of the membrane, although the receptors are concentrated in the flagellar

**eLife digest** African Trypanosomes are a group of single-celled parasites that are a major concern for livestock farmers in sub-Saharan Africa. They are carried by the tsetse fly and can cause disease in domestic livestock that diminishes productivity through reduced growth, and may ultimately lead to death. The parasites are coated in a dense layer of protein that help them evade the host's immune system by preventing immune cells from identifying them.

Humans have evolved immunity to many trypanosome species by exploiting a weakness in their lifestyle. Trypanosomes need to get haem—a molecule found in the protein haemoglobin—from their host to survive. In blood plasma, haemoglobin is found associated with a carrier protein called haptoglobin. To acquire haem, the parasites have a protein called HpHbR that binds to these haptoglobin-haemoglobin 'complexes'. However, in humans there are two complexes of proteins called TLFs that contain haemoglobin and a protein related to haptoglobin. The TLFs are also able to bind to HpHbR and are taken into the parasite. Once inside, TLFs cause internal compartments called lysosomes to swell, which leads to the death of the trypanosome.

Two subspecies of *Trypanosoma brucei* are the only trypanosomes that infect humans as they can overcome the TLF1 defense. However, the details of how TLFs cause cell death at the molecular level are not clear.

Lane-Serff et al. used a technique called x-ray crystallography to generate 3-D images of the HpHbR protein from *T. brucei* bound to the haptoglobin-haemoglobin complexes. These images show that HpHbR is elongated so that it only binds to haemoglobin and haptoglobin when they are together as a complex.

The images also reveal that the shape of HpHbR enables it to hold apart the proteins in the protective layer that coats the trypanosome. This allows the haptoglobin-haemoglobin complex to bind to HpHbR, but in humans also makes HpHbR more likely to bind to TLF1. These findings may help to guide future efforts to protect humans and livestock from the diseases caused by trypanosomes.

pocket, an invagination of the plasma membrane at the base of the flagellum and the site of all endocytosis (*Mussmann et al., 2004*; *Vanhollebeke et al., 2008*).

Humans, together with a few other primates, display innate immunity to most trypanosome species (*Laveran, 1902*) through the action of trypanolytic factors-1 and -2 (TLF1 and TLF2) (*Hager et al., 1994*; *Raper et al., 1996*, *1999*). Although containing different scaffold components, these factors both include apolipoprotein L1 (ApoL1) together with complexes of haemoglobin bound to haptoglobin-related protein (HprHb) (*Vanhamme et al., 2003*; *Pérez-Morga et al., 2005*). TLF1 enters trypanosomes via receptor-mediated endocytosis, through binding of the HprHb component to HpHbR (*Drain et al., 2001*; *Widener et al., 2007*; *Vanhollebeke et al., 2008*). This delivers ApoL1 to the endosome where it causes lysosomal swelling and cell death (*Pérez-Morga et al., 2005*). In contrast, the uptake route for TLF2 is unclear as, unlike TLF1, it is able to kill HpHbR null mutants (*Capewell et al., 2013*; *Uzureau et al., 2013*).

Just two subspecies of *T. brucei* (*T. b. rhodesiense* and *T. b. gambiense*) have evolved counter measures to the trypanolytic factors, allowing them to cause Human African Trypanosomiasis (*Pays et al., 2014*). In the case of human-infective group 1 *T. b. gambiense*, a unique point polymorphism is found in HpHbR (*Symula et al., 2012*) that reduces the monovalent affinity for ligand by 20-fold (*Higgins et al., 2013*). This contributes to resistance to TLF1, illustrating the importance of HpHbR.

Haptoglobin-haemoglobin is an elongated 'dumbell-shaped' complex consisting of a dimer of haptoglobin molecules, each joined to an αβ haemoglobin dimer (*Andersen et al., 2012*). Trypanosomes take up this HpHb complex but not the individual components (*Vanhollebeke et al., 2008*). The structure of the *T. congolense* HpHbR is an elongated three-helical bundle with a small membrane distal head (*Higgins et al., 2013*). Residues involved in HpHb binding are part of a small conserved patch ~25 Å below the tip of the receptor, but details of ligand binding and uptake were not characterized.

Here, we present the structure of *T. brucei* HpHbR. We show that the receptor adopts a similar architecture to its *T. congolense* homologue, but with a ~50° kink a third of the way along from the membrane proximal end. We also present the structure of TbHpHbR in complex with HpHb, revealing the molecular basis for ligand binding and selectivity. Finally, we show that the kink allows two independent membrane

attached receptors to interact with a single dimeric HpHb molecule and confirm using cell uptake experiments that this causes dimeric ligand to be taken up with greater efficiency than monomeric ligand. This reveals the molecular basis for the uptake of HpHb and trypanolytic factor-1 and identifies adaptations in the trypanosome receptor that allow efficient ligand uptake in the context of the tightly packed VSG coat.

## Results

### TbHpHbR binds to the HpSP domain:Hb head structure

To provide detailed molecular knowledge of the mechanism of uptake of haptoglobin-haemoglobin and trypanolytic factor-1 (TLF1), we aimed to determine the structure of *T. brucei* HpHbR (TbHpHbR) alone and bound to a human haptoglobin-haemoglobin complex.

TbHpHbR is longer than its homologue from *T. congolense* due to the presence of an additional C-terminal membrane-proximal domain. We therefore used the previously determined structure of *T. congolense* HpHbR (*Higgins et al., 2013*) to design a construct containing the corresponding region of TbHpHbR (residues 36–299). This region of the protein is identical in the human infective *T. b. rhodesiense*.

Haptoglobin-haemoglobin consists of a dimer of haptoglobin chains, each interacting with an αβ dimer of haemoglobin, and adopts a dimeric 'dumbell-shaped' architecture (*Andersen et al., 2012*). At each end, a serine protease (HpSP) domain of haptoglobin forms a stable complex with a haemoglobin dimer. Dimerisation occurs through an interface formed by the CCP domains of haptoglobin, linking together these HpSPHb 'heads'.

Previous studies have shown that TbHpHbR interacts with the HpHb complex but not with either haptoglobin or haemoglobin alone (*Vanhollebeke et al., 2008*), suggesting that that the receptor most likely binds to the heads of HpHb, where its two constituent components come together. We therefore designed a human haptoglobin construct containing just the SP domain (residues 148–406). This was expressed in baculovirus-infected insect cells and was combined with haemoglobin extracted from human blood to assemble HpSPHb complexes. We used surface plasmon resonance to determine the affinity of these HpSPHb complexes for TbHpHbR, and showed binding with an affinity of 0.7 µM (*Figure 1—figure supplement 1*), similar to the 1 µM affinity observed for intact human HpHb (*Higgins et al., 2013*).

Proteolytic cleavage of haptoglobin normally occurs in the endoplasmic reticulum after residue R102 but this cleavage event did not occur in the insect cell expressed HpSP domain. However, this did not affect the affinity for TbHpHbR. The shortened TbHpHbR construct and the HpSPHb complex therefore interact together with the same affinity as the full-length components, providing reagents for structural determination. These findings also confirm that TbHpHbR binds to the 'head' structure of dimeric HpHb, raising the possibility of two receptors simultaneously interacting with one HpHb complex.

### Determination of the structure of TbHpHbR alone and in complex with HpSPHb

To investigate the molecular basis for HpHb binding by TbHpHbR, crystallisation plates were set up for HpSPHb, TbHpHbR and a complex containing TbHpHbR bound to HpSPHb. Crystals of HpSPHb diffracted to 2.05 Å and were of space group P3$_1$21 with one complex in the asymmetric unit. Crystals of TbHpHbR diffracted to 1.85 Å resolution and were of space group P2$_1$ with two molecules in the asymmetric unit. Crystals of the TbHpHbR:HpSPHb complex were of space group C2 and diffracted to 3.1 Å resolution with a single complex in the asymmetric unit (*Table 1*).

The structure of human HpSPHb was determined using molecular replacement with the equivalent region of porcine HpHb (pdb: 4F4O) as a search model. The structure of the TbHpHbR:HpSPHb complex was then determined through molecular replacement using HpSPHb as a search model, allowing a poly-alanine model of TbHpHbR to be built. This model was then used as a molecular replacement search model to determine the structure of TbHpHbR using higher-resolution data obtained from crystals of the receptor alone. Both structures were then completed using iterative cycles of model building and refinement (*Table 2*).

### The structure of the *T. brucei* haptoglobin-haemoglobin receptor

Like *T. congolense* HpHbR, the *T. brucei* receptor is elongated, consisting primarily of a three-helical bundle (*Figure 1*): helix I (red; residues 42–110), helix II (orange; residues 116–182), and helix V (dark blue; residues 224–296) with a total length of 118 Å. At the membrane distal end, the receptor widens to form a compact head structure that includes the N-terminus and a 42-residue loop containing two

**Table 1.** Crystallographic data collection statistics

|  | HpSPHb | Tbb HpHbR | TbbHpHbR:HpSPHb |
|---|---|---|---|
| Beamline | Diamond I04-1 | Diamond I03 | Diamond I03 |
| Space Group | p3₁21 | p2₁ | c2 |
| Cell dimensions (Å) | a = b = 96.6, c = 132.77 | a = 27.90, b = 47.79, c = 203.38, β = 92.79 | a = 223.4, b = 56.59, c = 65.29, β = 92.99 |
| Resolution (Å) | 2.05 | 1.85 | 3.1 |
| Wavelength (Å) | 0.916 | 0.9763 | 0.9750 |
| $R_{PIM}$ (%) | 8.1 (37.4) | 4.5 (42.9) | 6.3 (72.6) |
| I/ σ(I) | 8.7 (2.3) | 10.2 (2.0) | 9.8 (1.6) |
| Completeness (%) | 99.8 (100) | 97.4 (96.5) | 96.9 (97.1) |
| Multiplicity | 9.6 (10.2) | 3.1 (3.1) | 3.2 (3.3) |

further helices, helix III (yellow: residues 186–196) and helix IV (green: residues 207—218). The upper part of the structure is extremely similar to that from *T. congolense*, with the membrane distal halves of the two receptors aligning with a root mean square deviation of 1.1 Å (*Figure 1—figure supplement 2*).

The most dramatic difference between the *T. brucei* and *T. congolense* receptors is a ~50° kink in TbHpHbR, located approximately one-third of the way along the receptor from the membrane proximal end. Each of the three helices is affected, with the backbone carbonyl groups of Asp88, Ala89, Glu123, Asn124, Asp270 and Ala271 no longer forming hydrogen bonds. This kink is not caused by flexibility, but is a rigid feature of the receptor, as it adopts the same confirmation in crystals of receptor alone, and in crystals of its complex with HpSPHb (*Figure 2A*), and is also observed in molecular envelopes derived from small angle x-ray scattering (*Figure 1*, *Table 3*). Instead it is caused by changes in the pattern of hydrophobic and hydrophilic residues around the kink site in each of the three helices. The three long helices of the *T. congolense* receptor are characterised by an alternating pattern of hydrophobic and hydrophilic residues, leading to continuous hydrophobic strips along the length of each helix that pack in the core of the helical bundle, stabilising its fold. In the *T. brucei* receptor, this pattern is disturbed at each kink site, breaking the organisation of the helix and leading to an alteration in the surface that displays the hydrophobic patch (*Figure 1C*). This stabilises the kink and makes it a rigid feature of the receptor structure.

## The structure of TbHpHbR in complex with haptoglobin-haemoglobin

The structure of the TbHpHbR:HpSPHb complex reveals an unexpected binding mode in which the ligand-binding surface extends along more than half of the length of the receptor (*Figure 2*,

**Table 2.** X-ray refinement statistics

| Complex | HpSPHb | Tbb HpHbR | TbbHpHbR:HpSPHb |
|---|---|---|---|
| Resolution (Å) | 2.05 | 1.85 | 3.1 |
| No. reflections | 43,170 | 44,685 | 17,302 |
| $R_{work}$ / $R_{free}$ (%) | 18.0 / 22.4 | 19.84 / 23.95 | 19.5 / 21.7 |
| No. of protein residues in model | 544 | 523 | 782 |
| rmsd bond lengths (Å) | 0.020 | 0.017 | 0.012 |
| rmsd bond angles (°) | 2.0 | 1.6 | 1.5 |
| Ramachandran plot |  |  |  |
| Allowed region | 89.0% | 98.8% | 92.5% |
| Additional allowed region | 11% | 1.2% | 7.5% |
| Generously allowed region | 0% | 0% | 0% |
| Disallowed region | 0% | 0% | 0% |

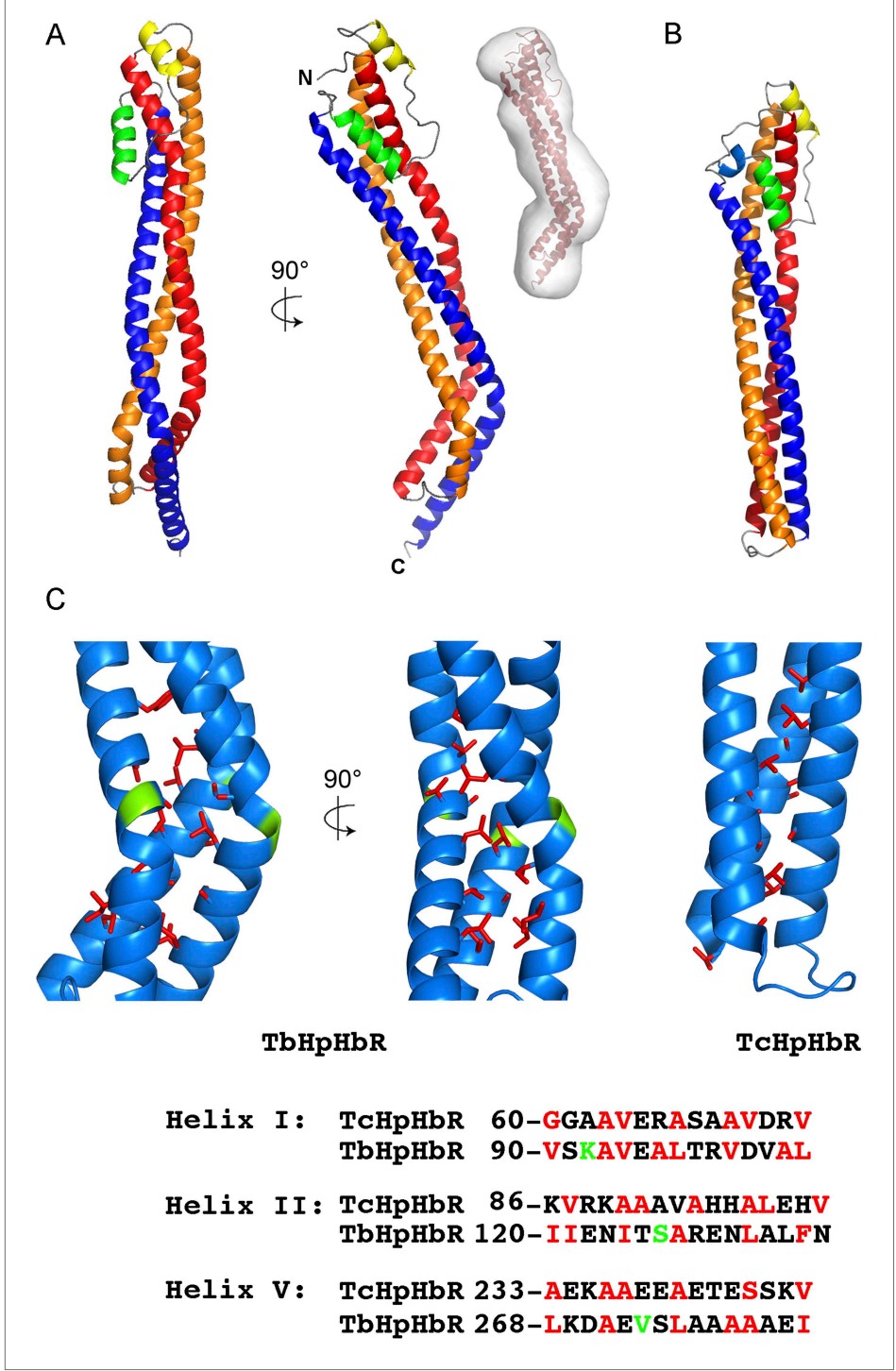

**Figure 1**. The structure of the *T. brucei* haptoglobin-haemoglobin receptor. (**A**) The structure of the *T. brucei* haptoglobin-haemoglobin receptor, with helix I (red), helix II (orange) and helix V (blue). These three helices form an elongated bundle with a ~50° kink towards the membrane proximal C-terminal end. The inset shows a molecular envelope derived from small angle x-ray scattering. (**B**) The structure of the *T. congolense* haptoglobin-haemo-globin receptor (**Higgins et al., 2013**) for comparison. (**C**) A change in the pattern of hydrophobic residues results in a rigid kink in the three helical bundle of the TbHpHbR. Corresponding regions of the structures of TbHpHbR and TcHpHbR are shown with side chains of the hydrophobic residues that pack in the core of the bundle coloured

*Figure 1. Continued on next page*

*Figure 1. Continued*

red and residues at the kink sites in TbHpHbR coloured green. Also shown are sequence alignments of TbHpHbR and TcHpHbR for these regions of each helix, coloured in the same way.

The following figure supplements are available for figure 1:

**Figure supplement 1**. Surface plasmon resonance analysis of the binding of HpSPHb to TbHpHbR.

**Figure supplement 2**. Alignment of the TbHpHbR and TcHpHbR structures.

*Figure 2—figure supplement 1*). Residues previously identified as playing a role in HpHb binding in TcHpHbR, such as S59 (*Higgins et al., 2013*), lie ~35 Å from the membrane distal tip of the receptor and directly contact haemoglobin. However, this is the upper part of the binding site, with residues from haptoglobin interacting as far as 70 Å from the membrane distal tip. This arrangement is confirmed by small angle x-ray scattering, with complexes of HpSPHb bound to either *T. brucei* or *T. congolense* receptors showing a similar architecture to that observed in the crystal (*Figure 2—figure supplement 2*, *Table 3*).

The haptoglobin-haemoglobin complex covers a total area of ~1250 Å$^2$ of the receptor and can be divided into two distinct regions (*Figure 2B*). The membrane distal part, (~745 Å$^2$) contacts the β-subunit of the haemoglobin dimer with no contacts between the receptor and the haemoglobin α-subunit. The membrane proximal region (~505 Å$^2$) forms a binding surface for haptoglobin. The involvement of both haemoglobin and haptoglobin in binding explains why the receptor binds HpHb but not haptoglobin alone. Modelling suggests that the lack of haemoglobin binding is due to steric clashes of the receptor with the second αβ dimer of haemoglobin when the β-subunit of a haemoglobin tetramer is docked onto the receptor with the binding mode observed in the TbHpHbR:HpSPHb complex (*Figure 2—figure supplement 3*). Therefore, the conformation of the receptor and the presence of two distinct binding sites allow the receptor to specifically select HpHb over its two constitutive components.

The haemoglobin β-subunit makes a number of direct interactions, mostly hydrogen bonds, with the receptor (*Figure 2C*, *Table 4*). Side chains from helix I of the receptor make the majority of these contacts, with additional interactions from helix II and the loop that links helices III and IV. These features lie along a groove on haemoglobin that is formed by helices C and F of the β-subunit. The haem group also makes direct contacts with the receptor, with the propionate chains contacting residues K56, S59, K164, R199 and Y200 of the receptor. These interactions, mediated by haem, form ~140 Å$^2$ of the ~745 Å$^2$ total contact area of Hb.

The haptoglobin subunit also interacts with helix I of the receptor, through a predominantly hydrophobic contact, mediated by three loops that emerge from the C-terminal β-sheet of haptoglobin (*Figure 2B*, *Table 4*). The structure of human haptoglobin from this complex aligns with that from porcine Hp with a root mean square deviation of just 0.5 Å and reveals no significant structural change on receptor binding (*Figure 2—figure supplement 4*). The alignment also confirms that the natural cleavage of Hp does not affect TbHpHbR binding, as residues in the loop that contains the cleavage site are not close to the receptor.

Rather than haptoglobin, trypanolytic factor-1 (TLF1) contains haptoglobin-related protein (Hpr) and binding of HprHb complex to TbHpHbR results in TLF1 uptake. The HprSP domain contains a total of sixteen amino acid substitutions when compared with the HpSP domain. Mapping these onto the structure shows that none of these differences lie in residues that contact the receptor (*Figure 3A*). Indeed HprSPHb complexes, prepared using the same protocols as HpSPHb complexes, bound to the receptor with an affinity of 1.7 µM, as determined by surface plasmon resonance (*Figure 3B*), comparable to the 0.7 µM affinity of the receptor for HpSPHb. This suggests that HprHb, and as a result, TLF1, will have a shared binding mode with HpHb.

## A model for haptoglobin-haemoglobin uptake in the context of the VSG layer

The haptoglobin-haemoglobin receptor operates in the context of the VSG layer, a dense coat of surface protein that covers the trypansosome surface. It is therefore initially surprising that the

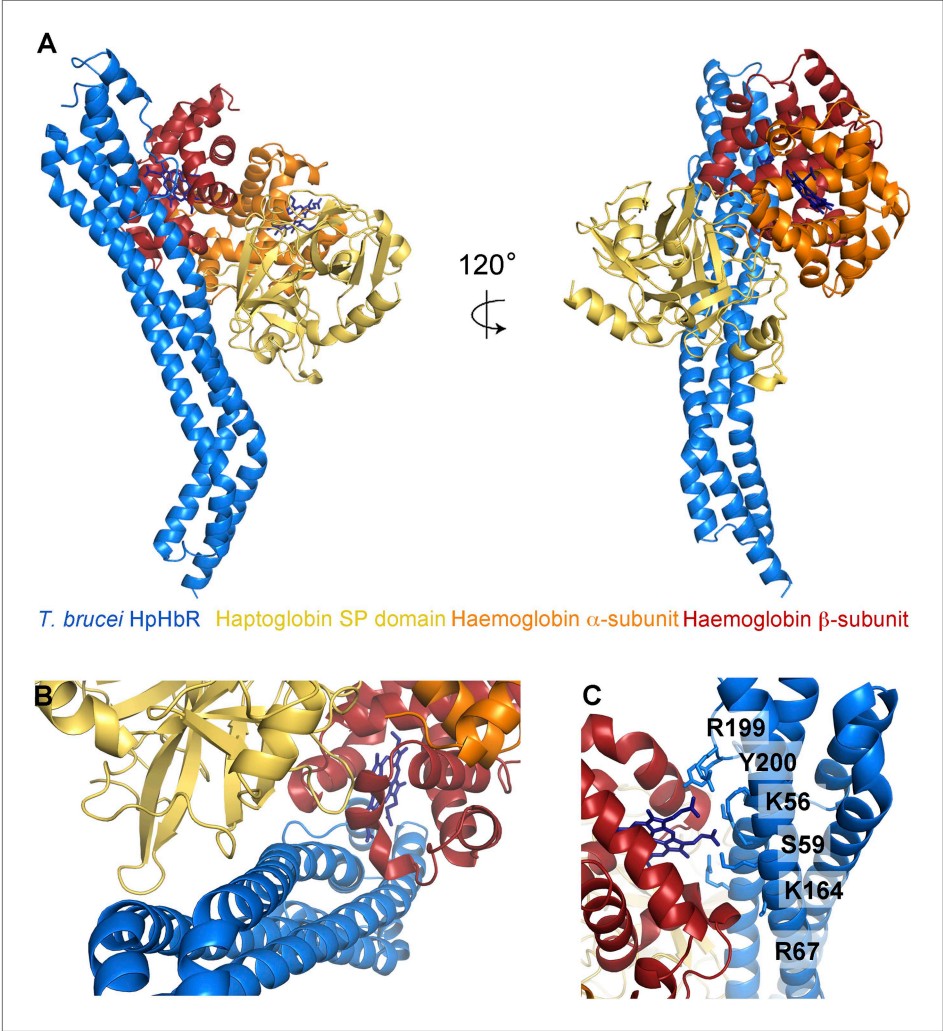

*T. brucei* HpHbR   Haptoglobin SP domain   Haemoglobin α-subunit   Haemoglobin β-subunit

**Figure 2**. The structural basis for haptoglobin-haemoglobin binding by TbHpHbR. (**A**) The structure of the complex between *T. brucei* HpHbR (blue) bound to its ligand, HpSPHb (haptoglobin is yellow, the β-subunit of haemoglobin is red and the α-subunit of haemoglobin is orange). (**B**) The complex viewed from the membrane proximal end, showing the contacts made by haptoglobin and the β-subunit of haemoglobin. (**C**) A view of the haemoglobin-binding site showing direct contacts between the haem and the receptor. Residues from the receptor that directly contact the haemoglobin subunit are shown as sticks and are numbered.

The following figure supplements are available for figure 2:

**Figure supplement 1**. Stereoview of the TbHpHbR in complex with HpHb.

**Figure supplement 2**. Small angle x-ray scattering of complexes of TcHpHbR and TbHpHbR with HpSPHb.

**Figure supplement 3**. Clashes between TbHpHbR and a haemoglobin tetramer explain why the receptor does not bind to haemoglobin.

**Figure supplement 4**. The region affected by haptoglobin cleavage is not involved in interaction with TbHpHbR.

location of the binding site for bulky HpHb complexes extends some 70 Å from the membrane distal tip of the receptor and below the surface of the VSG layer. However, one consequence of the kink in the *T. brucei* receptor is to increase its effective diameter, pushing apart VSG molecules. In addition, the orientation of the kink is precisely arranged to increase exposure of the HpHb binding site to the surface, making it more accessible for ligand binding.

**Table 3.** Small angle x-ray scattering statistics

| | MW (kDa) | $R_G$ (nm) | $D_{max}$ (nm) | Volume (nm³) | $Mw_{app}$ (kDa) |
|---|---|---|---|---|---|
| HpSPHb | 59.7 | 2.6 | 7.5 | 75 | 36 |
| TbHpHbR | 32.2 | 3.5 | 11.5 | 44 | 22 |
| TbHbHbR:HpSPHb | 91.8 | 3.2 | 10.8 | 110 | 55 |
| TbHpHbR:HpSPHb | 89.6 | 3.8 | 12.0 | 140 | 70 |
| HpHb | 152 | 5.6 | 18.2 | 214 | 107 |
| TbHpHbR:HpHb | 217 | 6.3 | 16.5 | 370 | 185 |

Docking of TbHpHbR:HpSPHb structures onto the structure of dimeric porcine HpHb reveals another consequence of the kink. This modelling suggests that two receptors can bind simultaneously to a single HpHb dimer, resulting in a C-shaped complex with a parallel arrangement of the membrane proximal parts of the two receptors (*Figure 4A*). Indeed, this arrangement was confirmed in solution by small angle x-ray scattering. Native (dimeric) human HpHb was mixed with TbHpHbR, and gel filtration was performed, with SAXS data collected from samples as they emerged from the column. The resultant scattering curves confirmed the assembly of a complex containing two receptors and one HpHb in vitro. These data were used to generate a molecular envelope for the complex, which confirmed the C-shaped architecture (*Figure 4B*, *Figure 4—figure supplement 1*, *Table 3*). Additional support for the formation of this complex in solution came from multi-angle laser light scattering (SEC-MALLS), which revealed masses of 30 kDa for the receptor, 150 kDa for HpHb and 210 kDa for the complex, showing that two receptors bind to each HpHb in solution (*Figure 4—figure supplement 2*). This arrangement, in which two

**Table 4.** Interactions between TbHpHbR and HpSPHb

| Receptor | | HpSPHb | | | |
|---|---|---|---|---|---|
| **Residue** | **Group** | **Chain** | **Residue** | **Group** | **Interaction** |
| | | Hbβ | | | |
| K56 | side chain | B | Haem144 | O1D | Hydrogen bond |
| E57 | side chain | B | K96 | Side chain | Salt bridge |
| S59 | side chain | B | Haem144 | O1D/O2D | Hydrogen bond |
| I60 | side chain | B | Patch | | Hydrophobic |
| R67 | side chain NH1 | B | R41 | Backbone CO | Hydrogen bond |
| E70 | side chain OE1/OE2 | B | R41 | Side chain NE/NH2 | Salt bridge |
| S161 | side chain | B | K60 | Side chain | Hydrogen bond |
| S161 | side chain | B | S45 | Backbone CO | Hydrogen bond |
| K164 | side chain | B | Haem144 | O2D | Hydrogen bond |
| R199 | side chain NE | B | Haem144 | O2A | Hydrogen bond |
| Y200 | side chain OH | B | Haem144 | O2A | Hydrogen bond |
| S203 | backbone CO | B | K96 | Side chain | Hydrogen bond |
| | | HpSP | | | |
| S73 | side chain | C | K345 | Side chain | Hydrogen bond |
| V74 | hydrophobic | C | Patch | | Hydrophobic |
| Q75 | OE1 | C | G276 | Backbone CO | Hydrogen bond |
| A78 | side chain | C | Patch | | Hydrophobic |
| A82 | side chain | C | Patch | | Hydrophobic |
| K85 | side chain | C | D305 | Side chain O2D | Salt bridge |

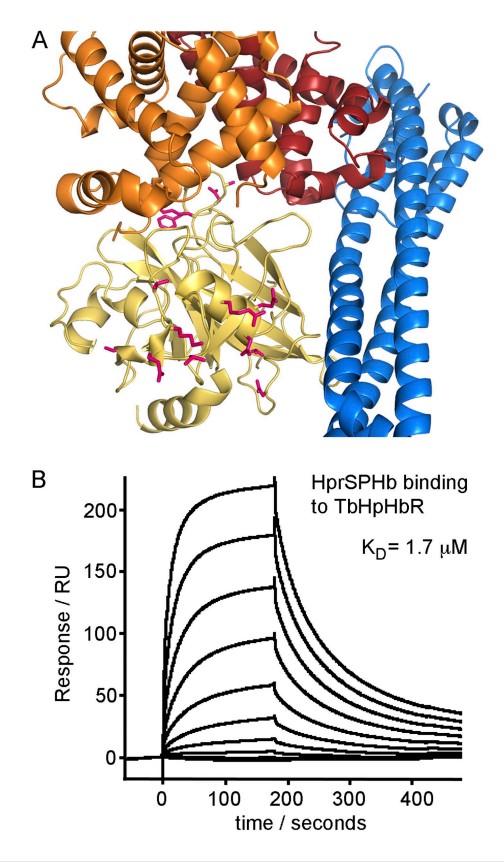

**Figure 3**. Differences between haptoglobin and haptoglobin-related protein do not alter affinity for TbHpHbR. (**A**) The structure of the TbHpHbR:HpSPHb complex is shown with the receptor in blue and haptoglobin in yellow. Side chains in haptoglobin that are different in haptoglobin-related protein are highlighted in pink and are not involved in making interactions with the receptor. (**B**) Surface plasmon resonance signals for two-fold dilutions of HprSPHb complex from a maximum concentration of 8 μM, binding to a surface coated with *T. brucei* HpHbR. The measured affinity of 1.7 μM can be compared with the affinity of 0.7 μM for HpSPHb.

independent GPI-anchored receptors can bind simultaneously to an HpHb dimer, would increase avidity for the ligand and decrease the ligand concentration required for efficient uptake.

To test this hypothesis, we performed uptake experiments using *T. brucei*. HpHb (dimeric), HpSPHb (monomeric) and bovine serum albumin (BSA, as a control to assess fluid phase uptake), were each fluorescently labelled. In addition, we prepared a null cell line, TbHpHbR$^{-/-}$, in which both copies of the receptor were disrupted (**Figure 4—figure supplement 3**). These reagents allowed us to investigate the concentration dependence of ligand uptake. In wild-type *T. brucei* cells, uptake of HpHb reached saturation below a concentration of 4 nM (**Figure 4C**). In contrast, the uptake of HpSPHb was negligible at a concentration of 4 nM and continued to increase at 62.5 nM (**Figure 4D**). Uptake of HpSPHb and HpHb observed in wild-type cells was due to the TbHpHbR, as expected (**Vanhollebeke et al., 2008**), as uptake of both ligands into TbHpHbR$^{-/-}$ cells was comparable to that of BSA in the range of ligand concentrations assayed (0–62.5 nM).

Therefore, uptake of dimeric HpHb into trypanosomes occurs efficiently at a significantly lower concentration than that of monomeric HpSPHb. As the monovalent affinities of the receptor for HpHb and HpSPHb are indistinguishable, as measured by surface plasmon resonance, this suggests that more efficient uptake of HpHb is caused by the dimeric ligand simultaneously binding to two receptors. Indeed, measurements of the binding of HpHb to immobilised receptor, using a very high surface density of HpHbR to measure bivalent binding, gave an affinity of 4.5 nM (**Higgins et al., 2013**), which is in the same range as the concentration at which HpHb uptake becomes saturated in live parasites. Therefore, it appears as though TbHpHbR has evolved a kink to increase accessibility of its ligand-binding site and to allow simultaneous binding of two receptors to one HpHb ligand, increasing ligand avidity and uptake efficiency.

## Discussion

The external surface of an African trypanosome is covered with a tightly packed layer of variant surface glycoprotein that shields epitopes that lie close to the plasma membrane from antibody binding (**Schwede et al., 2011**). Receptors such as those required for the uptake of transferrin and haptoglobin-haemoglobin complexes must operate within the context of this coat, with their structures organised such that ligand binding sites are not masked by the VSG layer.

Here, the first structure of a trypanosome receptor in complex with its ligand is presented: that of the *T. brucei* haptoglobin-haemoglobin receptor bound to haptoglobin-haemoglobin. Remarkably, the ligand-binding site extends more that half the way along the receptor, forming distinct binding surfaces for the β-subunit of haemoglobin and for haptoglobin. The simultaneous binding of both

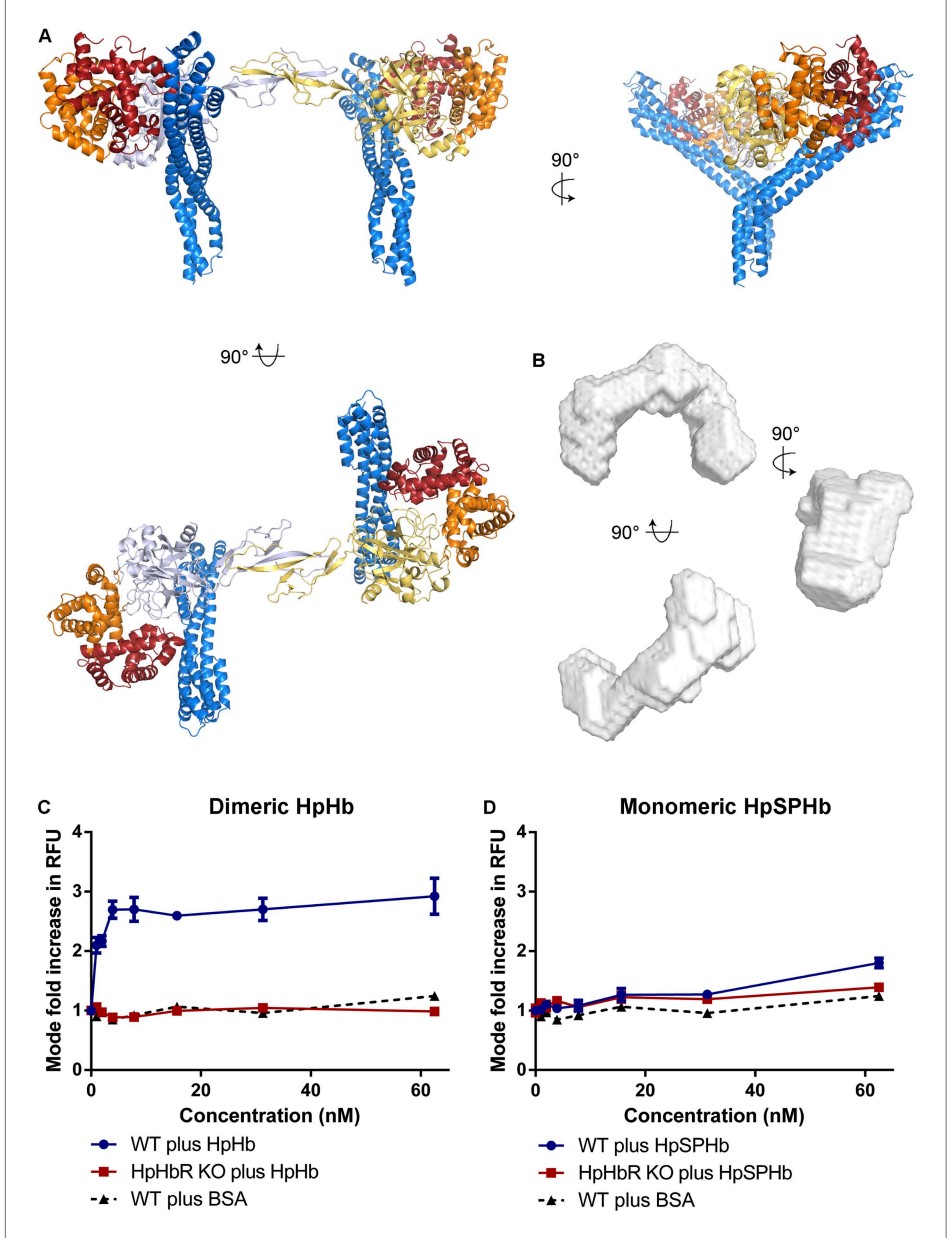

**Figure 4**. Simultaneous binding of two receptors to each HpHb dimer leads to more efficient uptake into trypano-somes. (**A**) A model for a complex of one HpHb dimer bound to two receptors, generated by docking the structure of the TbHpHbR:HpSPHb complex onto that of porcine HpHb (*Andersen et al., 2012*). The receptors are organized such that two receptors, both associated with the membrane through attachment at their C-termini, can simultane-ously bind to one HpHb dimer. (**B**) An ab initio molecular envelope derived from small angle x-ray scattering analysis of the TbHpHbR:HpHb complex supports the formation of a complex containing one HpHb dimer bound to two receptors. (**C**) Uptake of fluorescently labelled dimeric HpHb into live cells was monitored via flow cytometry across a range of 1–62.5 nM. Uptake saturated by 4 nM in wild-type cells whereas no uptake was observed in the HpHbR null cell line. No fluid phase uptake of labelled BSA was observed at these concentrations. (**D**) Uptake of fluorescently labelled monomeric HpSPHb was not readily detected until 62.5 nM, at which point uptake had not saturated. HpSPHb uptake at 62.5 nM was lost in the HpHbR null cell line. Each uptake assay was carried out in triplicate. Error bars represent standard error of the mean, n = 3.

The following figure supplements are available for figure 4:

**Figure supplement 1**. Small angle x-ray scattering of HpHb, alone and in complex with TbHpHbR.

*Figure 4. Continued on next page*

*Figure 4. Continued*

**Figure supplement 2**. SEC MALLS data to assess the stoichiometry of the TbHpHbR:HpHb complex.

**Figure supplement 3**. Establishment and characterization of an HpHb$^{-/-}$ cell line of *T. brucei*.

components explains the specificity of the receptor for haptoglobin-haemoglobin complexes over each individual component. However, the extent of the binding surface places it below the top of the VSG layer, apparently increasing the likelihood that it will be masked by VSG.

However, a ~50° rigid kink occurs as an adaptation in the three helical bundle of the *T. brucei* receptor, and we propose that it has two main functional consequences. Firstly, the direction of the kink is precisely arranged to bend the receptor such that the ligand-binding site becomes more exposed at the membrane surface. The kink will also increase the effective diameter of the receptor in the plane of the membrane. This combination is likely to increase the separation of VSG molecules in the region of the receptor and to increase the accessibility of the binding site for bulky macromolecular ligands such as HpHb and trypanolytic factors. Increased separation of VSG molecules by a trypanosome receptor is not a novel phenomena, with bulky glycan chains attached to the transferrin receptor proposed to have a similar effect (*Mehlert et al., 2012*), suggesting that different receptors increase the accessibility of binding sites for bulky ligands by different means.

The precise nature of the integration of TbHpHbR into the VSG layer remains unresolved as the effect of the C-terminal domains of both the receptor and VSG on the vertical disposition of each molecule remains unknown. An attractive model is that the ligand, whether HpHb or TLF1, is bound above the top of the VSG. However, the dimensions of the structures of VSG and the TbHpHbR:HpHb complex suggest that this may not be the case and that the HpHb ligand is held at least partially within the VSG layer (*Figure 5*). The TLF1 ligand is ~4 times the size of HpHb and this must, at least partially, protrude above the top of the VSG layer.

A second consequence of the kink is to allow two receptors to simultaneously bind to one dimeric HpHb complex when their membrane-proximal C-termini are membrane-attached. The affinity of a single receptor for HpHb is modest, at ~1 µM, with a rapid off-rate. By enabling two receptors to bind simultaneously, the kink will increase the ligand avidity, changing the effective affinity to something in the low nanomolar range, as previously measured for bivalent binding by TbHpHbR (*Higgins et al., 2013*) and to live cells (*Drain et al., 2001*).

What is the likelihood of two receptors interacting with a single ligand on the cell surface? The receptor copy number is 200–400 and it is concentrated in the flagellar pocket (*Vanhollebeke et al., 2008*). The surface area of the flagellar pocket in live cells is 4.3 µm$^2$ (*Grünfelder et al., 2002*). If there are 200 receptor molecules in the membrane of the flagellar pocket then the density is one receptor per 0.022 µm$^2$. The diffusion constant for HpHbR is unknown, but the diffusion constant for another GPI-anchored protein, VSG, has been measured by fluorescence recovery after photobleaching to be 0.01 µm$^2$/s (*Bulow et al., 1988*); this means that the receptor will contact a receptor molecule approximately every 2 s. The $t_{1/2}$ for release of monovalently bound HpHb is 70–100 s (*Higgins et al., 2013*) (*Figure 1—figure supplement 1*), so if it is assumed that the receptor has a similar diffusion coefficient to the VSG, then it is very likely that a monovalently bound ligand will become bivalently bound. Indeed our analysis of uptake of dimeric HpHb and monomeric HpSPHb into *T. brucei* confirmed that this increased avidity does occur in vivo, with HpHb uptake saturating at a concentration below 4 nM, while HpSPHb uptake is far from saturation at 62.5 nM.

Whether other trypanosome receptors use a similar avidity-increase mechanism to improve the efficiency of ligand uptake remains to be seen, and whether it is required will depend upon the receptor affinity for monomer and the sera concentration of the nutrient. However, it is clear from the example of TbHpHbR that this mode of ligand binding is potentially applicable to other GPI-anchored cell surface proteins.

While the evolution of the kink allows increased accessibility of the binding site for HpHb, it would also increase accessibility for the large TLF1 complex. One mechanism used by human infective *T. b. gambiense* to avoid TLF1-mediated innate immunity is a point polymorphism in HpHbR that reduces the monovalent affinity for HpHb by 20-fold (*Higgins et al., 2013*) and reduces TLF1 uptake (*Kieft et al., 2010*). It remains to be seen whether TLF1 contains one or multiple HprHb

**Figure 5**. A comparison of the dimensions of the TbHpHbR:HpHb complex with those of the N-terminal domains of the variant surface glycoproteins (shown in grey). This suggests that HpHb will lie at least partially within the VSG layer when bound to two receptors. DOI: 10.7554/eLife.05553.020

complexes, and whether these are in a suitable arrangement to allow bivalent binding. However, even if bivalent binding does occur in TLF1, the difference in affinity due to the *T. b. gambiense* polymorphism will be amplified under conditions where two receptors bind to a single ligand.

As the kink in the receptor appears to have a number of functional consequences that facilitate ligand uptake, it is surprising that the *T. congolense* receptor lacks such a kink. One reason for this difference might be the lack of a C-terminal domain in *T. congolense* surface proteins (*Higgins et al., 2013*). Perhaps the direct attachment of the ligand-binding domain to a GPI-anchor provides enough flexibility to allow the receptors to adopt an angle that allows simultaneous uptake, or perhaps the VSG coat of *T. congolense* parasites is less densely packed.

These questions will need further study.

In conclusion, we present the first structure of a trypanosome receptor in complex with its ligand and reveal a number of adaptations that are tailored to facilitate efficient ligand binding in the context of the VSG coat. These will decrease the packing of VSG molecules in the immediate vicinity of the receptor and increase accessibility of the ligand-binding site. They also allow two receptors to bind to a single ligand, thereby increasing avidity and dramatically decreasing the ligand concentration at which efficient uptake occurs. While different adaptations might facilitate each of these goals in different receptors, we would expect them to be general principles, frequently used by the parasite to aid nutrient uptake and survival.

## Materials and methods

### *T. brucei* HpHbR cloning, expression and purification

Full-length *T. b. brucei* HpHbR, without the N-terminal signal sequence and C-terminal GPI-anchor addition sequence, had been previously cloned for expression in a modified pET-15b to generate a polypeptide with an N-terminal hexahistidine tag and a cleavage site for TEV protease (*Higgins et al., 2013*). To produce a truncated construct for expression of the N-terminal ligand-binding domain, a stop codon was inserted after residue R299 using a polymerase chain reaction based mutagenesis protocol, using oligonucleotide GAGATGAAGCGCTAGGGGAACCCGATC and its reverse-complement. Mutagenesis was carried out as described for the Quikchange mutagenesis method (Stratagene, La Jolla, CA) and the plasmid was sequence verified.

The protein was expressed in *E. coli* Origami B, induced with 1 mM IPTG and incubated overnight at 18°C. The protein was purified by $Ni^{2+}$-NTA affinity chromatography and cleaved overnight with his-tagged TEV protease at 4°C in PBS with 3 mM oxidized glutathione, 0.3 mM reduced glutathione, followed by reverse $Ni^{2+}$-NTA affinity chromatography. The protein was concentrated by Amicon Ultra centrifugal filter device (10,000 MWCO) (EMD Millipore, Billerica, MA) and gel filtered using a Superdex 75 16/60 column (Life Technologies, Carlsbad, CA) into 20 mM HEPES pH 7.5, 150 mM NaCl.

### HpSP and HprSP cloning, expression and purification

Synthetic genes encoding the SP domains of human Hp (148–406) and Hpr (90–348) were cloned into a modified pAcGP67A vector to generate a polypeptide with an N-terminal hexahistidine tag and a cleavage site for TEV protease. These were transfected into Sf9 insect cells using the BaculoGold Baculovirus DNA transfection protocol (BD Biosciences, Franklin Lakes, NJ). Following selection of virus using plaque assays, the third amplification of recombinant virus was used to infect Sf9 insect cells. After 3 days, the cells were centrifuged for 15 min at 6000×*g*. After filtering, the supernatant was buffer exchanged into 20 mM Tris pH 8, 300 mM NaCl using a tangential flow apparatus (Pall Corporation, Port Washington, NY), followed by $Ni^{2+}$-NTA affinity chromatography. The protein was

concentrated using an Amicon Ultra centrifugal filter device (10,000 MWCO) (EMD Millipore). When used for crystallisation, HpSP was deglycosylated by incubation with endoglycosidase Hf (Sigma-Aldrich, St Louis, MO) and endoglycosidase F3 at enzyme:protein ratios of 1:25 in 1 mM $CaCl_2$, 1 mM $MgCl_2$, 100 mM HEPES pH 7.5 at 37°C for 3 hr.

## Purification of Hb and formation of the HpHb complex

Hb was isolated from human blood by sonication, followed by anion exchange chromatography using a Mono Q column (Life Technologies). HpHb was made by mixing full-length Hp 1-1 (Sigma-Aldrich) with purified Hb and isolating the complex by gel filtration using a Superdex 200 16/60 column (Life Technologies) in 20 mM HEPES pH 7.5, 150 mM NaCl.

## HpSPHb and HprSPHb complex formation

HpSP or HprSP at a threefold molar excess was mixed with Hb and diluted fivefold into 20 mM Tris pH 8, 500 mM NaCl, 15 mM imidazole. The complex was purified by $Ni^{2+}$-NTA affinity chromatography, washed using the dilution buffer, and eluted into PBS containing 200 mM imidazole. The complexes were then concentrated using an Amicon Ultra centrifugal filter device (10,000 MWCO) (EMD Millipore), and purified by gel filtration using a Superdex 200 16/60 column (Life Technologies) in 20 mM HEPES pH 7.5, 150 mM NaCl.

## Crystallisation, data collection and structure determination of the HpSPHb complex

HpSPHb was concentrated to 15 mg ml$^{-1}$ for crystallization. Crystals were obtained after 8 hr in sitting drops with a well solution containing 0.2 M NaCl, 0.1 M sodium cacodylate pH 6.5 and 2 M ammonium sulphate. These were cryoprotected by transfer into well solution with the addition of 30% vol/vol glycerol before cryo-cooling using liquid nitrogen. Data were collected on beamline I04-1 at the Diamond light source and were integrated and scaled using iMosflm (*Battye et al., 2011*) and scala (*Evans, 1993*) from the CCP4 suite (*Winn et al., 2011*), giving a final resolution of 2.05 Å. The structure was determined by molecular replacement using Phaser (*McCoy et al., 2007*) with the equivalent region of the porcine HpHb (pdb: 4F4O) as a search model. The structure was rebuilt and refined using Coot (*Emsley et al., 2010*) and REFMAC (*Murshudov et al., 2011*).

## Crystallisation, data collection and structure determination of the TbHpHbR:HpSPHb complex

TbHpHbR was mixed with HpSPHb and purified by gel filtration using a Superdex 200 16/600 column (Life Technologies) into a buffer containing 150 mM NaCl and 20 mM HEPES pH 7.5. It was concentrated to a final concentration of 15 mg ml$^{-1}$ and crystallised at 18 °C using sitting drops with a well solution containing 12.5% vol/vol MPD, 0.03 M NaBr, 0.03 M NaI, 0.03 M NaF, 0.1 M MES/imidazole pH 6.5, 12.5% wt/vol PEG 1000, 12.5% wt/vol PEG 3350 from the Morpheus screen (Molecular Dimensions, UK). Crystals formed after 10 days. Seed beads (Hampton Research, Aliso Viejo, CA) were used to create seeds from these crystals. These were used to seed a plate containing 100 nl of protein, 50 nl of the Morpheus well solution, and 50 nl of the Silver Bullet additive screen (Hampton Research). Crystals grew after 8 days in the well containing additives 0.2% wt/vol 2,2′-thiodiglcolic acid, 0.2% wt/vol apidic acid, 0.2% wt/vol benzoic acid, 0.2% wt/vol oxalic acid anhydrous, 0.2% wt/vol terephthalic acid. These were cryo-cooled in liquid nitrogen in the Morpheus well solution.

Data were collected on beamline I03 at the Diamond light source. Data reduction was performed using XDS (*Kabsch, 2010*) and the structure was solved by molecular replacement with the HpSPHb structure as a search model using Phaser (*McCoy et al., 2007*). Automatic model building in Buccaneer (*Cowtan, 2006*) was used to identify the positions of the receptor helices, leading to a cycle of model building and refinement in Coot (*Emsley et al., 2010*) and Buster (*Bricogne et al., 2011*). The coordinates from the higher resolution structures of both and TbHpHbR, also determined during this study, were used to provide restraints during refinement, leading to improved stereochemistry of the resultant model.

## Crystallisation, data collection and structure determination of TbHpHbR

The receptor was concentrated to 12.5 mg ml$^{-1}$ for crystallization. Crystals were obtained at 18 °C after 7 days using sitting drops with a well solution of 0.15 M KBr, 30% wt/vol PEG 2000 MME from

the JCSG+ screen (Molecular Dimensions). These were partially dehydrated and cryoprotected by transfer into the well condition with addition of 30% vol/vol glycerol before cryo-cooling in liquid nitrogen.

Data were collected on beamline I03 at the Diamond light source. Data reduction was performed using iMosflm (*Battye et al., 2011*) and scala (*Evans, 1993*) from the CCP4 data processing suite (*Winn et al., 2011*). Molecular replacement was performed using Phaser (*McCoy et al., 2007*), with the structure of TbHpHbR taken from the TbHpHbR:HpSPHb complex as a search model. A cycle of refinement and model building was carried out using REFMAC (*Murshudov et al., 2011*) and Coot (*Emsley et al., 2010*).

## Small-angle X-ray scattering (SAXS) data collection and processing

SAXS data for TbHpHbR alone and in complex with HpSPHb, and for the complex of TcHpHbR with HpSPHb, were collected at the PetraIII P12 beamline at Deutsches Elektronen-Synchrotron using a wavelength of 1.24 Å. SAXS data for the complex of the receptor with dimeric HpHb were collected at beamline BM29 at the European Synchrotron Radiation Facility using a wavelength of 0.9 Å. SAXS data for HpHb alone was collected at beamline B21 at the Diamond Light Source with a wavelength of 1.0 Å. In all cases, scattering was detected using a Pilatus image reader at 20°C.

The receptors alone and in complex with HpSPHb, as well as HpHb alone, were prepared at concentrations of 2.0, 1.0, 0.5, 0.25 and 0.125 mg ml$^{-1}$ in 20 mM HEPES pH 7.5, 150 mM NaCl. Twenty consecutive frames of 10 s each were recorded for each protein sample with a buffer sample measured between each, except HpHb, for which 180 consecutive 1 s frames were taken. Any images where the data had been affected by protein radiation damage were excluded from further processing.

The complex with dimeric HpHb was prepared and analysed by size-exclusion column (SEC)-SAXS, using a Superdex 200 10/300 column (Life Technologies) in 20 mM HEPES pH 7.5, 150 mM NaCl with a running speed of 0.4 ml min$^{-1}$. One frame was collected every 2 s. Frames corresponding to the peak seen on the UV trace were selected and a curve representing the scattering due to buffer was produced by averaging ten frames from the beginning of the run.

For each data set, PRIMUS (*Konarev et al., 2003*; *Petoukhov et al., 2007*) was used to normalize data to the intensity of the incident beam, for averaging of equivalent images and to subtract scattering due to buffer. Where Guinier plots revealed aggregation due to high concentration, data were removed (*Guinier and Fournet, 1955*). Composite curves were generated by scaling and merging the data sets.

AutoRg calculated the distance distribution function (P(r)) using an indirect Fourier transform, allowing estimation of the radius of gyration (R$_g$), the maximum particle dimension (D$_{max}$) and the Porod volume (*Porod, 1982*) by GNOM (*Petoukhov et al., 2007*). Initial models of the shape were generated using DAMMIF (*Franke and Svergun, 2009*) and averaged using the DAMAVER programme suite (*Konarev et al., 2006*). DAMMIN then produced a final model by minimising differences between experimental data and scattering of the model. The envelope model was produced using Situs (*Birmanns et al., 2011*), and feature-based docking of the crystal structures was completed using Sculptor (*Birmanns et al., 2011*).

## Surface plasmon resonance

Measurements were performed on a Biacore T200 (Life Technologies) instrument with a constant flow rate of 30 μg ml$^{-1}$. A CM5 chip was prepared by flowing over a 1:1 mixture of ethyl-dimethylaminopropyl-carbodiimide and N-hydroxysuccinimide. TbHpHbR was diluted into 10 mM sodium acetate pH 5 to a final concentration of 0.1 μM.

Ligands were diluted into HBS (20 mM HEPES pH 7.5, 150 mM NaCl, 0.005% vol/vol Tween 20). Both channels were equilibrated with HBS before injection of binding partner, and the level of specific binding obtained from a subtraction of the response from channel 2 from that of channel 1. Values for K$_D$ were obtained by equilibrium binding analysis using the BIAevaluation software.

## Size exclusion chromatography-multiangle laser light scattering

Purified samples were loaded onto a Superose 6 10/300 column (GE Healthcare), then analysed using laser light scattering detected at 662 nm wavelength at 8 scattering angles between 20.6° and

149.1° using a Heleos 8 instrument (Wyatt Technology, Germany). ASTRA 6.1 (Wyatt Technology) was used to calculate molecular weights using the Zimm equation. The samples were loaded at concentrations of 10 μM for HpHb and 40 μM for TbHpHbR.

## Trypanosome cell culture and construction of *T. brucei* HpHbR⁻/⁻ cell line

*T. brucei* blood stream form cells were grown in HMI-9 at 37 °C with 5% $CO_2$ (*Hirumi and Hirumi, 1989*). The linear DNAs used to replace HpHbR genes by homologous recombination were produced by PCR. First, one allele was replaced in procyclic form Lister 427 cells using a PCR product that contained 80 bp from upstream of the HpHbR gene, followed by the blasticin resistance cassette, followed by 80 bp from downstream of the HpHbR gene. The same approach was used with a G418 resistance cassette. Genomic DNA was prepared from blasticin or G418 resistant cell lines and used as a template for a PCR using oligonucleotides 500 bp upstream and downstream of the HpHbR gene. These second PCR products were used to serially transfect Lister 427 bloodstream form cells expressing VSG118.

## Uptake assays for monitoring uptake of fluorescently labelled ligands into live cells

When used for uptake experiments, Hp, HpSP and BSA were labelled with Alexa Fluor 488 using the protein labelling kit (Life Technologies). The manufacturer's protocol was adapted, extending the reaction time to overnight at 4 °C to increase labelling efficiency. Hp and HpSP were then subsequently mixed with Hb to form complex as above. For each assay, $5 \times 10^6$ wild type Lister 427 or *HpHbR* ⁻/⁻ cells were resuspended in 100 μl of serum free HMI-9 with 1% BSA and incubated with 2 μM protease inhibitor FMK-024 for 10 min at 37 °C. Cells were then incubated with 1–62.5 nM fluorescently labelled protein for 2 hr at 37 °C before being washed twice in serum free HMI-9 with 1% BSA. Cells were fixed in 4% paraformaldehyde for 10 min at room temperature and resupended in PBS. Uptake was assayed by flow cytometry using a FACScan (BD Biosciences) and quantified on FlowJo software. Mode increase in fluorescence was measured relative to a no ligand negative control, and all assays were carried out in triplicate.

## Acknowledgements

This work was supported by Medical Research Council Project Grant MR/L008246. HL is funded by the Wellcome Trust PhD program in Structural Biology. MKH is a Wellcome Investigator. We thank David Staunton for help with biophysical methods and Paul McKean (Lancaster University) for the gene deletion method. The structures are deposited in the protein data bank as 4X0I, 4X0J and 4X0L.

## Additional information

### Funding

| Funder | Grant reference number | Author |
| --- | --- | --- |
| Medical Research Council | MR/L008246/1 | Paula MacGregor, Edward D Lowe, Mark Carrington, Matthew K Higgins |
| Wellcome Trust | 101020/Z/13/Z | Harriet Lane-Serff, Matthew K Higgins |

The funders had no role in study design, data collection and interpretation, or the decision to submit the work for publication.

### Author contributions

HL-S, MKH, PMG, Conception and design, Acquisition of data, Analysis and interpretation of data, Drafting or revising the article; EDL, Acquisition of crystallographic data; MC, Conception and design, Analysis and interpretation of data, Drafting or revising the article

# Additional files

## Major datasets

The following datasets were generated:

| Author(s) | Year | Dataset title | Dataset ID and/or URL | Database, license, and accessibility information |
|---|---|---|---|---|
| Lane-Serff H, MacGregor P, Lowe ED, Carrington M, Higgins MK | 2014 | Trypanosoma brucei haptoglobin-haemoglobin receptor in complex with human haptoglobin-haemoglobin | http://www.pdb.org/pdb/explore/explore.do?structureId=4X0I | Publicly available at RCSB Protein Data Bank. |
| Lane-Serff H, MacGregor P, Lowe ED, Carrington M, Higgins MK | 2014 | Trypanosoma brucei haptoglobin-haemoglobin receptor | http://www.pdb.org/pdb/explore/explore.do?structureId=4X0J | Publicly available at RCSB Protein Data Bank. |
| Lane-Serff H, MacGregor P, Lowe ED, Carrington M, Higgins MK | 2014 | Human haptoglobin-haemoglobin complex | http://www.pdb.org/pdb/explore/explore.do?structureId=4X0L | Publicly available at RCSB Protein Data Bank. |

The following previously published dataset was used:

| Author(s) | Year | Dataset title | Dataset ID and/or URL | Database, license, and accessibility information |
|---|---|---|---|---|
| Andersen CB, Torvund-Jensen M, Nielsen MJ, de Oliveira CL, Hersleth HP, Andersen NH, Pedersen JS, Andersen GR, Moestrup SK | 2012 | Structure of the Haptoglobin–Haemoglobin Complex | http://www.pdb.org/pdb/explore/explore.do?structureId=4F4O | Publicly available at RCSB Protein Data Bank. |

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
