## [Decision Letter]

Thank you for sending your work entitled “Structural basis for ligand and innate
immunity factor uptake by the trypanosome haptoglobin-haemoglobin receptor” for
consideration at *eLife*. Your article has been favorably evaluated by
John Kuriyan (Senior editor) and 3 reviewers, one of whom is a member of our Board of
Reviewing Editors.

The following individuals responsible for the peer review of your submission have agreed
to reveal their identity: Volker Dötsch (Reviewing editor) and Michael Ferguson
(peer reviewer).

The Reviewing editor and the other reviewers discussed their comments before we reached
this decision, and the Reviewing editor has assembled the following comments to help you
prepare a revised submission.

This manuscript describes the structural investigation of the haptoglobin-haemoglobin
receptor (HpHbR) of *Trypanosoma brucei* in complex with human
haptoglobin-haemoglobin(HpHb) by x-ray crystallography as well as SAXS measurements.
This is an important interaction in trypanosome molecular biology, as it allows the
parasite to acquire haem, and it is relevant to the innate immune response humans have
to many species of trypanosome parasites. While the structures of the individual
components were largely known previously, the details of their complex were hitherto
unknown and are revealed for the first time here. The biological experiments,
demonstrating the different efficiencies of receptor-mediated uptake of monomeric and
dimeric HpHb ligands are an important component of the paper that nicely supports the
structure-based hypotheses on increased ligand avidity made possible by a 'two
receptors bind one ligand dimer' model.

Overall, the reviewers consider this a very nice paper showing important results. The
reviewers have not found any major concerns but have pointed out some minor issues and
questions that are summarized below.

Reviewer #2

1) I would suggest that the authors consider presenting a final figure of how they think
a receptor pair/ligand dimer would 'sit' in a sea of VSG molecules (top and
side view). This would be helpful for the reader, and I am sure would be popular as a
discussion and teaching aid.

2) While I like the receptor pair/ligand dimer model (particularly since it is supported
by uptake data) it leaves one quandary not discussed in the paper: how do two HpHbR
molecules find each other to bind the ligand dimer with high-avidity? The receptors are
GPI anchored (which helps) and so is the VSG coat so they presumably can wander the
surface but what is the probability of a 3-way collision (of 2 independent receptors and
a ligand dimer) on a 2-dimensional surface when the density of the receptors is low? Is
the model of transient single receptor-ligand dimer interactions that are occasionally
'locked down' when another receptor arrives by chance? Some discussion I think
is called for (but not additional experiments).

Reviewer #3

I am not an expert on small angle X-ray scattering, but it seems to me that the envelope
is a tight fit for the model in Figure 2 and a
generous fit in Figure 2. The authors might
comment on this. While these data are largely confirmatory, the SAXS data in Figure 4 are predictive, and there needs to be
confidence in the interpretations.

Many of the structure Figures (and especially Figure 2) would benefit from stereo views.

The paragraph “TLF1enters trypanosomes via receptor mediated
endocytosis…”, in the Introduction section, is confusing and does not
clearly explain what TLF1 and TLF2 are. The nomenclature in this paper (field) is quite
complex. I wonder if some sort of introductory figure could help.

The 50 degree kink is interesting and interpreted in terms of sequence variations. It
would be useful to add some comment about how main chain hydrogen bonding is
affected.

In the Results section (“The structure of TbHpHbR in complex with
haptoglobin-haemoglobin”), what does the 1250 Å^2^ refer to? Is it
the sum of the areas buried on the partner proteins or is it the interface area?

In the Hb field, the helices have always been denoted by letters, A–H in the case
of the β-chain, rather than numbers.

In Table 2, the refinement statistics for
TbbHpHbR:HpSPHb look outstanding for a structure at modest resolution especially
compared to the Tbb HpHbR structure. The authors should comment briefly on this.

---

## [Author Response]

Reviewer #2

*1) I would suggest that the authors consider presenting a final figure of how
they think a receptor pair/ligand dimer would 'sit' in a sea of VSG
molecules (top and side view). This would be helpful for the reader, and I am sure
would be popular as a discussion and teaching aid*.

We have been hesitant to produce a model showing the structure of the TbHpHbR: HpHb
complex in the context of the VSG layer mainly because of our lack of knowledge of the
relative sizes of the C-terminal domains of TbHpHbR and VSG. When TbHpHbR adopts a
tilted conformation on binding to HpHb it is approximately the same height as a VSG
molecule, making it likely that the HpHb will lie, at least partially, within the VSG
layer. We have now included a figure to show this similarity, but do not feel
comfortable to go further in proposing a detailed model.

*2) While I like the receptor pair/ligand dimer model (particularly since it is
supported by uptake data) it leaves one quandary not discussed in the paper: how do
two HpHbR molecules find each other to bind the ligand dimer with high-avidity? The
receptors are GPI anchored (which helps) and so is the VSG coat so they presumably
can wander the surface but what is the probability of a 3-way collision (of 2
independent receptors and a ligand dimer) on a 2-dimensional surface when the density
of the receptors is low? Is the model of transient single receptor-ligand dimer
interactions that are occasionally 'locked down' when another receptor
arrives by chance? Some discussion I think is called for (but not additional
experiments)*.

The point made about the likelihood of two receptors colliding with a single HpHb is an
interesting one. Assuming that the receptors are predominantly localised at the
flagellar pocket, and using the known receptor copy number and VSG diffusion rates, we
have calculated that each receptor should contact another receptor approximately once
each second. As the t_1/2_ for the dissociation of HpHb from a single receptor
is in the region of 70-100 s, there is a strong likelihood that an HpHb attached to a
single receptor will interact with a second receptor before it dissociates, leading to
higher avidity bivalent binding. We have added a paragraph to the Discussion section to
make this point.

Reviewer #3

*I am not an expert on small angle X-ray scattering, but it seems to me that the
envelope is a tight fit for the model in*
Figure 2
*and a generous fit in*
Figure 2*. The authors
might comment on this. While these data are largely confirmatory, the SAXS data
in*
Figure 4
*are predictive, and there needs to be confidence in the
interpretations*.

We agree with the reviewer that it is essential to be confident about the interpretation
of the data supporting the formation of a complex containing two receptors bound to a
single HpHb in solution. We have made two changes to the manuscript to present data that
strengthens this conclusion. Firstly, we now include SAXS data for HpHb alone, allowing
comparison with that for the HpHbR:receptor complex. The envelopes derived from these
data, together with the accompanying molecular weights, support the binding of two
receptors to each HpHb. In addition, we have included SEC-MALLS data, which also shows
the formation of a complex containing two receptors bound to a single HpHb. These
biophysical data support the conclusions derived from crystallography and explain the in
vivo effects observed in HpHb uptake experiments.

*Many of the structure Figures (and especially*
Figure 2*) would benefit
from stereo views*.

We have added a new figure supplement for Figure 2 to show a stereoview of the TbHpHbR:HpSPHb complex.

*The paragraph “TLF1enters trypanosomes via receptor mediated
endocytosis…”, in the Introduction section, is confusing and does not
clearly explain what TLF1 and TLF2 are. The nomenclature in this paper (field) is
quite complex. I wonder if some sort of introductory figure could help*.

We have clarified the paragraph on the trypanolytic factors in the Introduction,
simplifying our description of these (imperfectly characterised) factors and focusing
our discussion onto the key ApoLI and Hpr components. We have also rearranged the
subsequent two paragraphs. We believe that this will improve their clarity to a reader
from outside the field and give a balanced view of what we know of the role of the HpHbR
in mediating innate immunity.

*The 50 degree kink is interesting and interpreted in terms of sequence
variations. It would be useful to add some comment about how main chain hydrogen
bonding is affected*.

We have added a sentence to the description of the kink in TbHpHbR in which we clarify
the effect on main chain hydrogen bonding, and we describe which residues in each of the
three helices have their hydrogen bonding disrupted.

*In the Results section (“The structure of TbHpHbR in complex with
haptoglobin-haemoglobin”), what does the 1250
Å*^*2*^
*refer to? Is it the sum of the areas buried on the partner proteins or is it the
interface area*?

The area of 1250 Å^2^ is the interface area, i.e. the area on the receptor
that is covered by HpHb, and not the sum of the buried surface areas on both proteins.
We have clarified this in the text.

*In the Hb field, the helices have always been denoted by letters, A–H in
the case of the β-chain, rather than numbers*.

We have corrected the labelling of the haemoglobin β-subunit helices, with
helices C and F indicated as those that contact the receptor.

*In*
Table 2*, the
refinement statistics for TbbHpHbR:HpSPHb look outstanding for a structure at modest
resolution especially compared to the Tbb HpHbR structure. The authors should comment
briefly on this*.

We agree with the reviewer that the refinement statistics for the TbHbHpR:HpSPHb
structure are very good. This is, in part, due to the use of additional restraints from
the higher resolution structures of the receptor and the HpSPHb complex during
refinement. As neither structure showed any conformational change upon complex
formation, these additional restraints were used to improve the stereochemistry of the
refined model. We were careful, by comparison of the final maps with those obtained
without use of these restraints, to show that their use did not lead to major changes in
the model. We have increased our explanation of our refinement procedure in the
Materials and methods to clarify this.

In addition, we have deposited the pdb codes and diffraction data to the protein data
bank with codes 4X0I, 4X0J and 4X0L.